# Novel Autoantibody Signatures in Sera of Patients with Pancreatic Cancer, Chronic Pancreatitis and Autoimmune Pancreatitis: A Protein Microarray Profiling Approach

**DOI:** 10.3390/ijms21072403

**Published:** 2020-03-31

**Authors:** Sahar Ghassem-Zadeh, Katrin Hufnagel, Andrea Bauer, Jean-Louis Frossard, Masaru Yoshida, Hiromu Kutsumi, Hans Acha-Orbea, Matthias Neulinger-Muñoz, Johannes Vey, Christoph Eckert, Oliver Strobel, Jörg D. Hoheisel, Klaus Felix

**Affiliations:** 1Department of General, Visceral and Transplantation Surgery, University Hospital Heidelberg, 69120 Heidelberg, Germany; sghassem@me.com (S.G.-Z.); neulingermatthias@gmail.com (M.N.-M.); oliver.strobel@med.uni-heidelberg.de (O.S.); 2Department of Biochemistry, University of Lausanne, 1066 Epalinges-Lausanne, Switzerland; Hans.Acha-Orbea@unil.ch; 3Infections and Cancer Epidemiology, German Cancer Research Center (DKFZ), 69120 Heidelberg, Germany; k.hufnagel@Dkfz-Heidelberg.de; 4Department of Functional Genomics, DKFZ, 69120 Heidelberg, Germany; andrea.bauer@dkfz-heidelberg.de (A.B.); j.hoheisel@dkfz.de (J.D.H.); 5Department of Medical Specialties, Division of Gastroenterology, University Hospital of Geneva, 1205 Geneva, Switzerland; jean-louis.frossard@hcuge.ch; 6Department of Gastroenterology, Kobe University Graduate School of Medicine, Kobe 650-0017, Japan; myoshida@med.kobe-u.ac.jp; 7Center for Clinical Research and Advanced Medicine Shiga University of Medical Science Seta Tsukinowa-cho, Otsu 520-2192, Japan; hkutsumi@belle.shiga-med.ac.jp; 8Institute of Medical Biometry and Informatics, University Medical Center Ruprecht-Karls University Heidelberg, 69120 Heidelberg, Germany; vey@imbi.uni-heidelberg.de; 9Institute of Pathology, University Hospital Heidelberg, 69120 Heidelberg, Germany; christoph.eckert@med.uni-heidelberg.de

**Keywords:** pancreatic cancer, chronic pancreatitis, autoimmune pancreatitis type 1 and type 2, antibodies, microarray protein

## Abstract

Identification of disease-associated autoantibodies is of high importance. Their assessment could complement current diagnostic modalities and assist the clinical management of patients. We aimed at developing and validating high-throughput protein microarrays able to screen patients’ sera to determine disease-specific autoantibody-signatures for pancreatic cancer (PDAC), chronic pancreatitis (CP), autoimmune pancreatitis and their subtypes (AIP-1 and AIP-2). In-house manufactured microarrays were used for autoantibody-profiling of IgG-enriched preoperative sera from PDAC-, CP-, AIP-1-, AIP-2-, other gastrointestinal disease (GID) patients and healthy controls. As a top-down strategy, three different fluorescence detection-based protein-microarrays were used: large with 6400, intermediate with 345, and small with 36 full-length human recombinant proteins. Large-scale analysis revealed 89 PDAC, 98 CP and 104 AIP immunogenic antigens. Narrowing the selection to 29 autoantigens using pooled sera first and individual sera afterwards allowed a discrimination of CP and AIP from PDAC. For validation, predictive models based on the identified antigens were generated which enabled discrimination between PDAC and AIP-1 or AIP-2 yielded high AUC values of 0.940 and 0.925, respectively. A new repertoire of autoantigens was identified and their assembly as a multiplex test will provide a fast and cost-effective tool for differential diagnosis of pancreatic diseases with high clinical relevance.

## 1. Introduction

Investigation tools to diagnose pancreatic ductal adenocarcinoma (PDAC) are of unsatisfactory sensitivity and specificity. Indeed, around 10% of patients that undergo surgery for suspected PDAC have benign inflammatory diseases, mainly chronic (CP) or autoimmune pancreatitis (AIP) [1,2]. However, these benign diseases do not usually require surgical resection except for compromised quality of life by major pain symptoms, deterioration of nutritional status or inability to determine the exact nature of the lesion. Moreover, AIP can be treated by steroids where a high response to corticosteroid therapy is an important diagnostic criterion. The distinction between AIP and particularly AIP-type 2 and PDAC can be challenging [1,2,3]. An international multicentre survey conducted in 2011, showed that 60% (123 of 204) and 78% (50 of 64) of respectively AIP-1 and AIP-2 were evaluated retrospectively from resected pancreas on suspicion of pancreatic cancer [4]. The need of new diagnostic tools is crucial to decrease these numbers. Therefore, an accurate diagnosis may pre-empt the misdiagnosis of cancer, allowing the appropriate medical treatment of AIP and consequently decrease the number of unnecessary pancreatic resections. 

Few tumour-associated antigens are established in clinical routine as serological markers for discrimination of PDAC and the most frequently used is the carbohydrate antigen CA 19-9 [5]. Although CA 19-9 is not sufficient enough for the screening of PDAC, it helps for the differential diagnosis between PDAC and pancreatitis, to assess treatment response, prognosis and follow-up of PDAC. Furthermore, Lewis antigen-negative individuals do not secrete CA 19-9 or secrete it in small amounts [6]. 

A distinguishing feature of autoimmune diseases and cancer is the expression of disease-associated autoantibodies. Their release into the blood circulation and their assessment could aid early diagnosis of high-risk populations and assist the clinical management of patients. Tumour associated autoantibodies (TAAbs) are promising serum biomarkers for detection of early stage disease [7,8,9,10,11]. Indeed, TAAbs can be detected earlier before the disease progresses to an advanced, incurable stage. The mechanisms by which autoantibodies are produced accompanying the development of cancer are complex and poorly understood. The immune system induces immunologic processes causing autoantibody production in response to mutations, overexpression of proteins, altered antigen folding, aberrant degradation, aberrant glycosylation and/or the release of proteins from damaged tissue [12,13]. Another explanation is that B cells generate autoantibodies (AAbs) by escaping the self-tolerance checkpoints of the immune system [14]. AAb may be detectable both at early onset of the disease and in higher concentrations as compared with the tumour antigen itself. Their long presence and synthesis due to limited proteolysis and clearance of the tumour antigen representing an in vivo amplification of signal detection [15]. As biomarkers, AAbs are highly specific and easily purified from serum which makes them an interesting tool for diagnosis and distinction between different disease groups. High-throughput screenings on larger sample cohorts, including patients with well-defined early stage diseases, are required to differentiate autoantibodies’ expression in malignant and benign pancreatic diseases and define disease specific signatures. Protein microarrays are modern high-throughput tools applicable to detect new disease-specific antigen signatures recognized by AAbs for diagnosis and prognosis, but also to help gain more insight into the molecular nature of diseased conditions [16]. 

Applying antigen microarray technology [17] we describe here the array manufacture process and subsequent analysis procedures leading to the identification of disease specific AAbs through profiling of preoperative sera derived from patients with resectable PDACs, CP, both AIP types, other GI-tract diseases and healthy controls. 

## 2. Results

### 2.1. Screening of Pooled IgG Enriched Sera Fractions on Large Protein Microarrays

The initial screening was performed on 16 large scale arrays with the first sample set composed of 60 serum samples assembled in four groups (PDAC-, CP-, AIP- patients and Co). Each group as a pool made of 15 individual sera was profiled on four arrays covering 6400 antigens each. Representative scans of four identical arrays profiled for the PDAC, CP, AIP and Co group are shown in Figure 1A.

The fold-change of PDAC, CP and AIP fluorescence intensity for each protein was evaluated. The median of all disease groups (PDAC, CP and AIP) arrays were normalized according to the mean of all array spots of the healthy control group (Co). The normalized median fluorescence intensity (MFI) of each antigen recognized/detected in the three different pools of patients’ sera (PDAC, CP and AIP) was compared to the MFI of the antigen recognized by the healthy Co pool. The antigens were selected as promising candidates when the normalized log-values revealed a fold change greater than 1.5 for PDAC, CP and AIP compared to Co. This allowed the identification of 425 autoantigens discriminating between disease-specific and disease-overlapping autoantibodies (Figure 1B). The candidates that were assigned to PDAC (*n* = 89), assigned to CP (*n* = 98) and to AIP (*n* = 104), and candidates overlapping CP and AIP (*n* = 54) were selected for the manufacture of intermediate-sized microarrays with 345 antigens and subjected to profiling with a larger cohort the sample set 2. 

Of note, only 28 autoantigens were found overlapping between the three disease groups PDAC, CP and AIP. The list of selected candidate proteins is summarized in Appendix A. 

### 2.2. Screening of Multiple Sera of a Large Cohort 

For the first refinement the above-mentioned intermediate-sized microarrays were used. On these 260 IgG enriched fractions were profiled as 52 pools of five patients each (pools: Co *n* = 14, PDAC *n* = 8, CP *n* = 7, GID *n* = 10 AIP-1 *n* = 10, and AIP-2 *n* = 3). Patients’ data are presented in Table 1 under “sample set 2”. After scanning the microarrays, an MFI was assigned to all proteins. Internal positive controls (EBV VCA p18) and negative controls (PBS) were used on each slide for quality control. All values were converted into log values. T-test was applied to compare all sets of observations and all *p* values were adjusted using false discovery rate (FDR) method. Out of 345 candidates, 29 were differentially recognized among the different disease groups. Identification of the 29 autoantigens is shown in Table 2. 

The ID correctness was confirmed by sequencing the PCR products, from which the relevant proteins were produced recombinantly. Of note, four of these proteins—annexin A4 (ANXA4), protease serine 1 (PRSS1), lactotransferrin (LTF) and peroxiredoxin 4 (PRDX 4)—were already previously reported as AIP autoantigens [7,18]. The normalized median log2 MFI for each candidate for the six compared groups are reported in Table 3.

In order to visualize the differential abundance of the 29 AAb between the 52 analysed pools, heat maps were generated (Figure 2A). Comparing all 29 candidates’ abundance in CP and its autoimmune variants AIP-1 and AIP-2 versus PDAC clearly segregates the benign from malign pancreatic diseases as presented in Figure 2B and demonstrates the lower immunogenic response in the PDAC group confirming an earlier microarray profiling report by Gnjatic et al [8].

### 2.3. Autoantibody Profiling of Selected Autoantigens with Individual Samples 

Using the profiling results of pooled sera on intermediate-sized arrays, the 29 selected candidates were spotted on new slides together with seven additional autoantigens for AIP (ANXA1, ANXA2, CA-I, CA-II, ENO1, YWHA and 14-3-3 ζ) repeatedly reported in the literature, by that forming a 36 spot array. Hence, a small array was formed comprising internal positive (EBV VCA p18) and negative (PBS) controls spotted in several replicates. For the analysis, the sample set number 3 (Table 1) consisting of 185 individual patients (Co *n* = 48, PDAC *n* = 25, CP *n* = 24, GID *n* = 26, AIP-1 *n* = 47, AIP-2 *n* = 15) was profiled. All groups were compared using ANOVA and Tukey’s multiple comparison tests. Initially, comparison of autoantigens between PDAC and all AIP patients (AIP-1 and AIP-2 together) was performed. Table 4 displays 10 antigens that significantly differ between these two groups. 

Nine of them revealed significantly higher median MFI in AIP than in PDAC sera. Surprisingly CA-II, representing a well-known autoantigen in AIP (*p* = 0.0173), was recognised significantly more in PDAC compared to AIP sera. Subsequently, all antigens were separately compared among all groups of patients’ sera. The selected candidates with corresponding median MFI and *p* values and for each disease group that were significantly different among the disease groups are presented in Appendix A. 

Among the antigens that discriminated between PDAC and benign inflammatory disease (Figure 3), three showed significantly different MFI between both AIP subtypes: PP1R15A (*p* = 0.0387), CYP3A5 (*p* = 0.0173) and WDR45 (*p* = 0.0173). Other antigens that do not discriminate PDAC from AIP but were found to discriminate between both AIP subtypes were MAGEA2 (*p* = 0.0306), FCGR2B (*p* = 0.0272) and WDR45 (*p* = 0.0280). Furthermore, three out of seven autoantigens that were previously reported to be associated with AIP—ANXA2 (*p* = 0.0173), ANXA4 (*p* = 0.0280) and ENO1 (*p* = 0.0280)—reacted significantly more with AIP-1 sera. 

### 2.4. Validation of the Marker Panel

In order to assess the validity of the 29 identified autoantigens regularized logistic regression using an elastic net was applied to build statistical models for the discrimination between, first, benign (AIP-1 + AIP-2 + CP) vs. malignant (PDAC), second, AIP-1 vs. PDAC and, third, AIP-2 vs. PDAC. For each classification two prediction models were built, one based on the selected 29 antigens (Table 2) and the other incorporating the additional seven antigens above mentioned and previously reported as AIP antigens. The variable importance was measured based on the mean absolute values of the regression coefficients estimated during cross-validation (upper graphs in Figure 4). For the discrimination between CP+AIP-1+AIP-2 and PDAC, nearly all antigens received mean absolute coefficients unequal to zero. To discriminate between AIP-2 and PDAC only 13 of the 29 antigens showed calculated mean absolute coefficients unequal zero (Appendix A). Notable is that often the same candidates, such as PPP1R15A, EIF2S2, PAICS, TOR1B, PRSS1, LENG1 and WDR45 were selected and showed large coefficients. The predictive performance of the models was quantified using ROC curves (Figure 4). The curves illustrate that achieving high sensitivity is only possible at the expense of specificity, and vice versa. The models based on the 29 and 36 antigens, respectively, only showed slight differences in their accuracy. 

The models discriminating between PDAC and AIP-1 and AIP-2, respectively, yielded high AUC values of 0.940 and 0.925, respectively. 

## 3. Discussion

Interest on antibodies as biomarkers for both autoimmunity and cancer has been a subject of debate these last decades [20,21]. Profiling on protein microarrays is one of the most powerful techniques that allow large-scale quantitative protein determination in a high-throughput way for initial protein biomarker discovery. The majority of autoantibody profiling investigations determined antigens recognized by autoreactive IgG antibodies. To this end they have been used to identify biomarkers for cancer diseases [22,23,24]. Microarrays assembled of thousands of different proteins have been already reported and used to identify tumour associated autoantibodies in the context of colorectal cancer [25], bladder cancer [26], ovarian cancer, and pancreatic cancer [8,27]. Furthermore, several studies described elevated autoantibody concentrations in sera of patients with AIP including anti-lactoferrin [28,29], anti-carbonic anhydrase II, anti-carbonic anhydrase IV, anti-pancreas secretory trypsin inhibitor, anti-anionic and cationic trypsinogens, anti-amylase-1, anti-heat shock protein 10 and anti-plasminogen-binding protein peptide autoantibodies [7,18,30]. However, their performance as diagnostic markers is still under debate. Some of these antibodies are organ-specific mainly against antigens from the pancreatic ducts and acini [28,30], whereas others are not organ-specific and directed against nuclear antigen (ANA). Recently, additional autoantibodies have been identified against pancreatic enzyme precursors in AIP patients [7]. The study also reported different expression of anti-pancreatic lipase and anti-transaldolase in the two AIP-subtypes.

Though AIP have been clearly recognized as two entities with distinct clinical profiles the aetiology and the pathophysiological mechanisms of AIP remain still unknown [4,31,32,33,34]. 

In the current study we aimed at the identification of new biomarkers able to discriminate between very similar pancreatic diseases. We showed the usefulness of a custom protein microarray approach that provides specific serum antibodies pattern in patients affected by PDAC, CP, AIP and GID. Starting with 6400 human recombinant proteins with subsequent two steps refinement strategy as selection, we identified 29 autoantigens which were differentially and significantly recognized by autoantibodies present in the IgG-enriched fraction sera of patients with pancreatic diseases.

Our study cohort was able to address the three following unsolved issues: (i) can autoantibodies differentiate between benign pancreatic diseases from PDAC, (ii) can autoantibodies differentiate AIP-patients particularly those with AIP type 2 from PDAC-patients and (iii) can autoantibodies differentiate both subtypes of AIP. We identified TOR1B, RNF138, PPP1R15A, PAICS, LENG1, GPR3 and CYP3A5 as autoantigens that allowed discrimination between PDAC, CP and AIP. 

Many previously described proteins used in the clinic to distinguish PDAC from healthy patient actually fail to discriminate PDAC from pancreatitis (CP and AIP), a spectrum of diseases that shares many molecular and imaging features with PDAC [35]. In these patients, CT scan can be non-diagnostic and more invasive endoscopic testing may be required toward a final diagnosis. A non-invasive pancreatitis-biomarker panel may be helpful for the investigation of such patients. From a clinical standpoint, the actual utility of a biomarker would depend on the context in which it is used. Indeed, biomarkers for PDAC necessarily need to have a very high specificity because of the low prevalence of the disease (0.01%) that would lead to numerous false positive. CA 19-9 is the most extensively studied and validated serum biomarker for the diagnosis of PDAC. With an overall sensitivity and specificity in the range of 80–86% both values do not reach the necessary sensitivity and specificity for pancreatic cancer detection. This unsatisfactory performance is partly due to elevated CA19-9 levels in benign diseases such as acute and chronic pancreatitis, biliary obstruction, cholangitis, and liver cirrhosis but also in other gastrointestinal cancers resulting in a significant number of false positive. Additionally in patients with Lewis negative genotype (5–10% of population) CA 19-9 is not expressed leading to false negative results.

On the other hand, the prevalence of CP, including autoimmune subtype, approaches 1–5% of the population and in such a population less specificity could be tolerated as long as the biomarker test can distinguish healthy controls. 

Some of the markers reported in this study are able to distinguish PDAC from AIP-1 but fail to discriminate between PDAC and AIP-2. This finding may be related to the fact that the clinical spectrum of AIP-1 is a systemic disease compared to AIP-2. Furthermore, it is difficult to distinguish AIP-2 from PDAC as both can induce the same clinical profile. We therefore assessed different antigen profiles between both types of AIP and PDAC and were able to generate predictive models with high AUC values for AIP1 and AIP2 vs. PDAC. 

The most interesting biomarker is Torsin 1B (TOR1B) because it is capable to discriminate AIP from other diseases. Torsin Family 1, member B also known as TOR1b or DQ1 is found primarily in the endoplasmic reticulum and nuclear envelope and can act as a chaperone allowing the maintenance of the integrity of the nuclear envelope and endoplasmic reticulum. 

Surprisingly only a small number of antigens have been identified that differentiate PDAC from other diseases. We can extrapolate that it could be related to the desmoplastic reaction that often takes place in this tumour [8]. Furthermore, heterogeneity among our patients with PDAC (stage and grade) could also explain variation of recognition of antigens. Nevertheless, the protein microarray screening was able to identify candidate proteins that show strong differential recognition between the pancreas diseases PDAC, CP and both AIP forms. 

We took a step-by-step analysis approach with three different but interlocking profiling parts. The third part involved analysis of a large cohort of patients tested individually. MFI were used to address statistical analysis. The statistical prediction models revealed high predictive performance with AUC values that reached about 90%. This leads us to conclude that the identified antigens might serve as a basis to establish accurate predictive models. However, valid models need to be calculated including a much larger sample size. The uncertainty of the predictive strength can be recognized by the moderate-sized confidence intervals.

Not only might the novel autoantibody panel help to improve diagnostics and to understand the pathophysiology of AIP, but the autoantibodies could also lead to new possible immunotherapeutic targets. Customized, small-sized arrays or similar assay formats could be a useful and affordable way to diagnose and discriminate pancreatic cancer from benign inflammatory diseases and avoid unnecessary surgery on patients with AIP-1 and AIP-2. 

## 4. Materials and Methods 

### 4.1. Patients and Samples

Serum samples of patients with pathologically confirmed PDAC, CP, AIP-1, AIP-2, gastro-intestinal diseases (GID) and controls were obtained from the Pancobank of the

European Pancreas Center (EPZ/Department of Surgery, University Hospital Heidelberg; Ethical Approval Votes no. 301/2001 and 159/2002), a member of BMBH/Biomaterial Bank Heidelberg. Additional sets of serum samples were provided from the Department of Gastroenterology, Kobe University Graduate School of Medicine, Kobe, Japan. 

Data of patients who were referred for an operation were collected in a prospectively designed database. The clinico-pathological parameters included age, gender, TNM classification and AJCC stage of tumour location are listed in Table 5.

The first group, the PDAC sera (65 patients mean age 66.2 ± 11 years) derived from patients with operable pancreatic tumours. The second investigated group, chronic pancreatitis (CP) consisted of 50 serum samples from patients who have undergone duodenum preserving pancreatic head resection (mean age 53.3 years ± 9.6). The histopathological analysis of all patients confirmed CP. The third group, represented by 55 AIP-1 patients, is a cohort composed of 25 and 30 serum samples from Heidelberg and Kobe, respectively (mean age 64.5 ± 12.2 years). The fourth group, AIP-2 consisted of 15 patients from Heidelberg only. All AIP cases were confirmed using histology whenever tissues were available. If histology was not available, the HISORt Mayo Clinic criteria or the International Consensus Diagnostic Criteria (ICDC) for AIP were used to confirm the disease. The fifth patient group was composed of benign and malign extra-pancreatic gastrointestinal diseases (GID), and consisted of 60 patients, (mean age 63.3 ± 10.8 years).As controls, a group of 70 healthy volunteers (Co) (mean age 47 ± 19.5 years) including 25 healthy controls from Kobe was used.

These six groups were used in three dependent sample sets in this study. Sample set 1 consisted of 60 pooled sera IgG fractions (15 sera/group), sample set 2 consisted of 260 sera pooled in 52 IgG fractions (5 sera/pool) and sample set 3 was composed of 185 individual patient sera. Exact numbers of the sera and the patients’ clinical data in each sample set used for controls, gastrointestinal diseases (GID), chronic pancreatitis (CP), autoimmune pancreatitis (AIP-1 and AIP-2) and pancreatic adenocarcinoma (PDAC) cohorts are presented in the Table 1, text and figures. 

### 4.2. IgG Enrichment

To limit artifacts in the array binding process and increase specificity, serum samples were enriched for IgGs using the NAb protein A/G spin column and buffers (ThermoScientific, Rockford, IL). The purification procedure was performed according to the manufacture’s protocol. The collected fractions were neutralized by adding 40 µL of 1.0 M Tris, pH 8.8, measured at 280 nm to assess the amount of purified antibodies and subsequently used for profiling procedures. 

### 4.3. Generation of DNA Templates for Recombinant Human Antigens from Bacterial Library

The ORF-clone library of the ORFeome Collaboration was provided by the DKFZ Genomic and Proteomic Core Facility [36]. Clones were selected according to the encoded protein’s role in inflammation and/or cancer. Two microliters of each selected clone were transferred with 150 µL of LB-kanamycin-medium and incubated at 37 °C overnight. The day after, the cultures were spun down at 1900× *g* for 30 min. The pellets were resuspended in 100 µL of PCR-grade water, heated in a ventilated oven at 75 °C for 20 min and spun down once more as above-described. Subsequently, the supernatants containing plasmid DNA were transferred into standard 96-well PCR plates. DNA-samples were stored at −20 °C. Five microliters of the isolated plasmid DNA used as a template for the synthesis of the gene of interest was amplified using PCR with a Taq DNA polymerase kit (Qiagen, Hilden, Germany). The kit was used according to the manufacturer’s instructions manual during 40 cycles of denaturation at 94 °C for 30 sec, annealing at 52 °C for 30 sec and elongation at 72 °C for 210 sec. This allowed the gene of interest to be amplified with a pair of expression primers carrying regulatory sequences and fusion tags: forward expression primer: 5’-GAAATTAATA CGACTCACTA TAGGGAGACC ACAACGGTTT CCCTCTAGAA ATAATTTTGT TTAAGAAGGA GATATACATA TGCATCATCA TCATCATCAT AAAGCAGGCT CCACCATG-3’; reverse expression primer: 5’-CTGGAATTCG CCCTTTTATT ACGTAGAATC GAGACCGAGG AGAGGGTTAG GGATAGGCTT ACCAACTTTG TACAAGAAAG CTGGGTC-3’. The C-terminal V5 sequence was intended for the detection of full-length expressed proteins. Agarose gel (1.3%) electrophoresis was performed to verify the amplified DNA fragments for correct DNA bp-length. The identity of the last 29 candidate autoantigens was further confirmed by sequencing using a Mix2seq kit (Eurofins). 

### 4.4. Manufacture of Protein Microarrays

The fabrication of protein microarrays was performed applying the multiple spotting technique, an approach that uses DNA templates to synthesize proteins directly on microarray slides [17,37]. The protein microarrays were spotted on epoxysilane coated slides (SCHOTT Nexterion AG, Jena, Germany) using a microarray non-contact printer, the Nanoplotter 2 (GeSIM, Radeberg, Germany) following the protocol described by Hufnagel et al. [38]. The protein microarrays manufacture process is outlined in Figure 5. Briefly, during a first spotting step 900 pL of each DNA template was transferred onto the epoxysilane slides. On top of each DNA spot, in a second spotting event, droplets were printed consisting of 3.6 nL of the cell-free expression mixture (S30 T7 High-Yield Protein Expression kit, Promega). Protein synthesis occurred during an incubation phase at 37 °C for one hour followed by an overnight incubation (12-16 hours) at 30 °C. Each candidate was spotted in duplicates. In addition, an experimental positive control (Epstein–Barr Virus VCA p18) and negative control consisting of PBS in PCR mixture were printed in several replicates on each array. 

All antigens corresponded to distinct human recombinant proteins. A quality control of the spotting procedure was performed randomly on 10% of each batch of production. The percentage of proteins that were successfully expressed on the protein microarray was assessed using fluorescence-conjugated antibodies directed against fusion tags (6xHis and V5) present at either end of the expressed proteins. Briefly, the slides were blocked using a 2% BSA buffer for one hour. All incubations were performed on a shaker at room temperature. After two washing steps with PBST, the Penta-His Alexa Fluor 647 conjugate antibody (Qiagen) and the anti-V5 Cy3 monoclonal antibody (Sigma) were incubated at a dilution of 1:1000 both in 2% BSA for one hour. The slides were then washed three times and dried in an oven at 30 °C prior to scanning using the Powerscan (Tecan, Männedorf, Switzerland) at two wavelengths (532 and 635 nm). The signal intensity was considered to be representative of the amount of expressed proteins on the slides. 

### 4.5. Immunoassay with Patient Antibodies 

In order to compare the reactivity of each antigen equally in the different patients, all IgG fractions used for profiling on microarrays were adjusted to a protein concentration of 10 µg/ml prior to incubation on the slides. As first step, the slides were blocked using SuperBlock (ThermoScientifc) for one hour at room temperature on a shaker. After two washing steps with PBST, 10 μg of IgG fractions diluted in PBS with 0.81 mg of *E*. *coli* lysate were incubated for 15 h to allow the antibodies to bind to the recombinant antigens on the microarray. Since an *E*. *coli*-based expression mixture was used for protein synthesis, remaining *E*. *coli* proteins on the slide could otherwise react with antibodies against *E*. *coli* proteins in the serum and thereby produce unspecific signals. After two more washing steps with PBST, a fluorescence-conjugated secondary antibody (goat anti-human whole IgG, (Jackson ImmunoResearch, Europe Ltd., Cambridgeshire, UK) diluted 1:200 in PBS directed against the primary antibodies from the patient sera on the microarray was added for two hours. The slides were afterwards washed three times for 15 min, rinsed in Milli-Q sterile water and dried at 30 °C. All slides were scanned (Powerscan, Tecan, Männedorf, Switzerland) using a 635 nm laser. The signal intensity obtained from the fluorescence-conjugated antibody was approximately proportional to the amount of primary antibody in the enriched-IgG sera that recognize the antigens on the protein microarray.

### 4.6. Construction and Optimization of the Protein Microarrays

Applying the top-down strategy as a stepwise refinement for selection of the best differentiating autoantigens, three differently sized protein microarrays were manufactured.

The large scale arrays (*n* = 16, four per profiled group, each with 1600 individual antigens/array covering a total of 6400 antigens) were printed and incubated with a small cohort of total 60 serum samples assembled in four groups of PDAC, CP, AIP patients and Co. Each group consisted of 15 individual sera (20 µL) combined in a pool (300 µL) and subjected to IgG-enrichment. The patients’ characteristics in this sample set 1 are presented in Table 1. 

The intermediate size microarrays, *n* = 60, consisted of 345 antigens spotted in duplicates/array. On these 260 IgG-enriched fractions were analysed: Co: *n* = 70; PDAC: *n* = 40; GID: *n* = 50; CP: *n* = 35 AIP-1: *n* = 50; AIP-2: *n* = 15; they were profiled as 52 pools of 5 patients per group (pools: Co *n* = 14, PDAC *n* = 8, CP *n* = 7, GID *n* = 10 AIP-1 *n* = 1 0, and AIP-2 *n* = 3). 

Finally, small-size arrays of 36 antigens in duplicates were printed and tested in sample-set 3 on cohorts of 185 patients individual IgG-enriched fractions (PDAC: *n* = 25; CP: *n* = 24; AIP-1: *n* = 47; AIP-2: *n* = 15; GID: *n* = 26; and Co: *n* = 48). 

### 4.7. Data Analysis

GenePix Pro software (Molecular Devices, Sunnyvale, CA, USA) was used to analyse the fluorescence intensity of each spot and to eliminate spatial artifacts. Several negative controls consisting of PBS in PCR mixture spotted on each array were used to assess possible nonspecific signals. 

After analysing the microarrays using the GenePixPro Software, the signal intensity expressed in median fluorescence intensity values (MFI), was normalized by linear scaling of log-ratios for all array probes. As variables were not normally distributed, the nonparametric two-sided Kruskal–Wallis test was used to assess differences of an antigen between disease groups. All *p* values were adjusted using false discovery rate (FDR) method. Adjusted *p*-values of < 0.05 were considered to be significant. 

Logistic regularized regression using an elastic net [39] was applied to build a prediction model based on selected antigens. The elastic net hyper-parameters α (elastic net mixing parameter) and λ (shrinkage parameter) were tuned by conducting a five-fold cross-validation, assessing variable importance maximize the AUC. An advantage of cross-validation is that each observation is assigned to the test data exactly once. Hence, for each patient the estimated probability for its classification can be extracted. Model building was performed with R version 3.6.2 [40] using the packages “glmnet” and “caret‘ [41,42]. ROC curves were plotted with the “pROC” package [43]. Confidence intervals for the AUC values were computed by the formula according to Wilson [19].

All graphs except Figure 5 were designed in GraphPad Prism software (version 5; La Jolla, CA, USA). The normalized MFI of each candidate were graphically presented as box-and-whisker plots. 

## 5. Conclusions

This work aimed at the identification of antigen signatures in patients with pancreatic cancer, a devastating disease often diagnosed at late stage and misdiagnosed with benign inflammatory diseases, such as CP and AIP. The medical needs for such biomarkers are not only important for diagnostic and discrimination but may also help to identify new possible immunotherapeutic targets. Indeed, the clinical diagnose of PDAC requires different investigation tools such as tissue sampling through CT scan or endoscopic fine needle aspiration, as well as serological markers (CA19-9 and CEA), that have several limitations [44]. Both markers are not only of poor sensitivity but also of poor specificity among other gastrointestinal diseases. To our knowledge this is the first study comparing serum from such an important and diverse cohort. The distinction between chronic and autoimmune pancreatitis is also important in terms of medical treatment and long-term outcome. Though, chronic pancreatitis has diverse etiological treatments, AIP requires in the vast majority of cases a long-term and specific corticoid-therapy. Furthermore, erroneous administration of corticoids to patients suffering from PDAC can not only exacerbate the cancer in terms of tumour invasion but also greatly contribute to delay the diagnosis of a very lethal disease. Our results revealed not only panels of biomarkers useful to discriminate between PDAC, CP and AIP but also markers able to distinguish between both types of AIP. This recognition pattern detected via custom protein array is a cheap, non-invasive, high specific test that should be used routinely in the clinic. The functional role of these antibodies is poorly described in the literature. Further experiments should address the role on the biological function of autoantibodies present in the IgG-fraction of patients that are targeting these antigens. Regardless of the heterogeneity of the antigens among the patient groups, these candidates could provide useful insight in mechanistic pathways and therapeutic intervention. Statistical modelling revealed potential to establish a predictive model based on the identified antigens for the differential diagnosis of pancreatic diseases. However, for this purpose, a larger sample size is required and the models need to be validated in a prospective manner. 

## Figures and Tables

**Figure 1 ijms-21-02403-f001:**
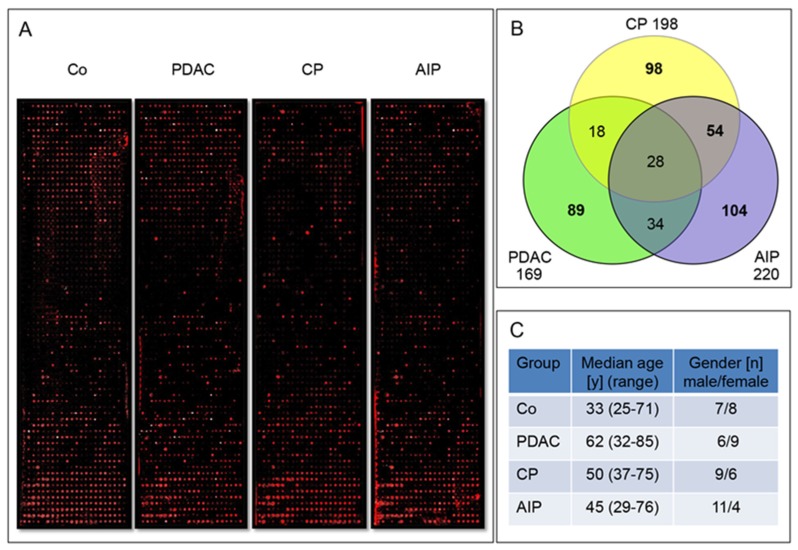
(**A**) Example of four microarrays equally spotted with 1600 human recombinant proteins each after incubation with enriched IgG serum fractions from healthy Co, PDAC, CP and AIP. Red dots indicate specific autoantigen/autoantibody complex formation and colour intensity represents the amount of the specific autoantibody bound. White dots indicate saturated signals with maximal intensity. For technical reasons, background signals were higher at the top and bottom end of the arrays. Still, the selection criteria permitted identification of real positive signals. (**B**) Number of autoantigens retrieved from the first sample set. The profiling allowed discrimination between disease-specific and disease-overlapping autoantibodies for PDAC, CP, and AIP. (**C**) Sample set 1 patients’ data was composed of 60 serum samples assembled in four groups.

**Figure 2 ijms-21-02403-f002:**
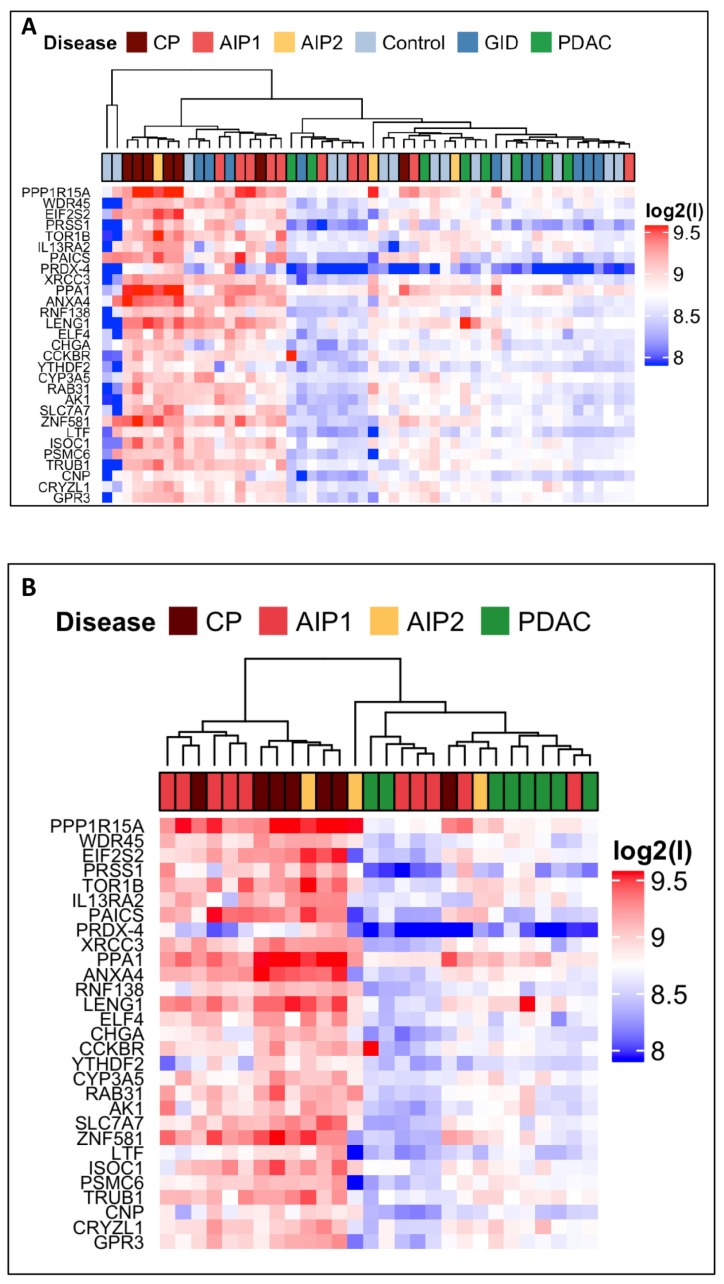
Overview of pooled serum antibody reactivity to the selected autoantigens. AIP-1, AIP-2 and CP mostly show higher autoantibody reactivity than PDAC and, thus, help to differentiate PDAC from primarily inflammatory pancreatic diseases. Heatmaps generated from intermediate-sized protein microarrays data presenting serum antibodies reactivity with the indicated antigens in the six tested cohorts. High autoantibody reactivity values are presented by red, low autoantibody reactivity by blue boxes. Data are presented as differential median fluorescence intensity MFI (MFI log2 scale) levels of the response to the 29 autoantigens. (**A**) For comparison of all cohorts the following pools were used: Co (*n* = 14); PDAC (*n* = 8); CP (*n* = 7); GID (*n* = 10); AIP-1 (*n* = 10), and AIP-2 (*n* = 3). (**B**) Hierarchical clustering using the log2 MFI of the 29 autoantigens between the three different pancreatic diseases and visualization of higher reactivity of CP- and both AIP subtypes-serum antibodies versus PDAC-serum antibodies.

**Figure 3 ijms-21-02403-f003:**
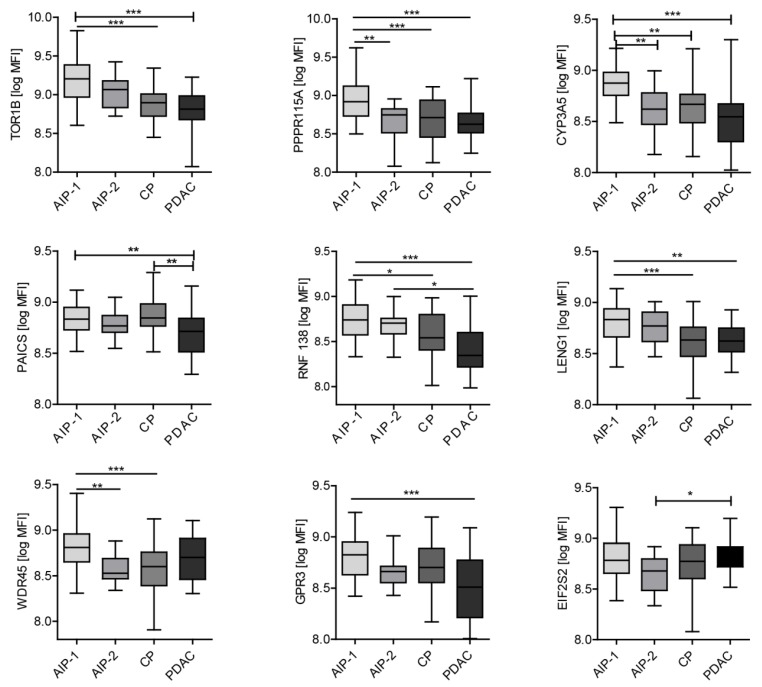
Comparison of antibody reactivity for serum antibodies (MFI) to autoantigens between the different pancreatic diseases. Differences were considered statistically significant when the *p*-value was less than 0.05 and are marked with an asterisk: * *p* < 0.05, ** *p* < 0.01 and *** *p* < 0.001.

**Figure 4 ijms-21-02403-f004:**
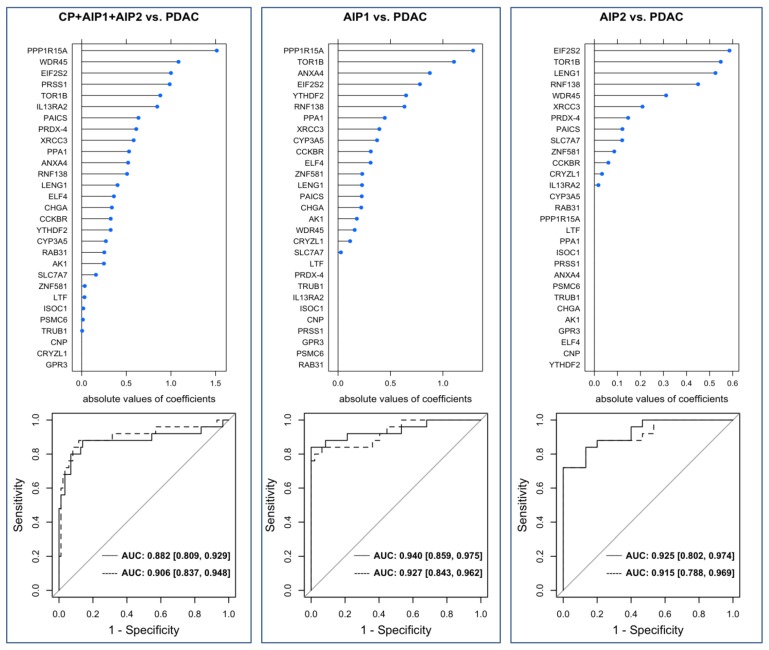
The predictive performance of the models. Upper graphs present the variable importance measured as mean absolute values of the coefficients estimated during cross-validation of the models incorporating the 29 identified antigens. Below, corresponding ROC curves of the statistical models discriminating between AIP-1+AIP-2+CP vs. PDAC (**left**), AIP-1 vs. PDAC (**middle**), and AIP-2 vs. PDAC (**right**). The ROC curves for the models based on the 29 antigens are depicted as continuous lines and the ROC curves for the models based on the 36 antigens as dashed lines. The corresponding confidence intervals of the AUC values are indicated in the square brackets. Confidence intervals for the AUC values were computed by the formula according to Wilson [19].

**Figure 5 ijms-21-02403-f005:**
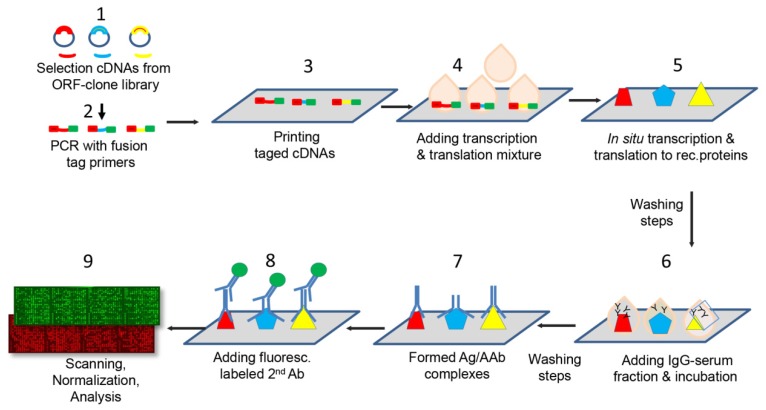
Simplified flow chart of the nine major steps for the manufacture of protein microarrays and their application for autoantibody detection and profiling. **1:** cDNAs selection from library **2:** Two-step PCR for enrichment cDNAs and fusion with tag primers. **3:** Spotting of cDNAs on the epoxisilane-coated slides. **4:** Addition of expression mixture to each DNA spot. **5:** Incubation of the DNA/expression mixture for cell free transcription and translation. Removal of unbound proteins by washing. The recombinant proteins were retained through an epoxy-amino reaction. **6:** Addition of IgG serum fractions and formation of antigen/antibody complexes. **7:** Removal of unbound IgGs. **8:** Incubation with fluorescence-conjugated secondary antibodies (anti-human IgG, IgM and IgA) directed against the primary antibodies from the patient sera. **9:** Scanning of microarrays using a 635 nm laser (assessment of fluorescence intensity of each spot and normalization and microarray data analysis.

**Table 1 ijms-21-02403-t001:** Patient data in the three sample sets used for protein microarray profiling, *n*: number of observations.

Patients Type	Sample Set	Patients (*n*)	Median Age (Range)	Male (*n*) (%)
**Co**	1	15	33.0 (25–71)	7 (46.7)
	2	70	40.5 (20–83)	29 (70.0)
	3	48	48.5 (20–81)	31 (64.6)
**PDAC**	1	15	66.0 (54–77)	6 (40.0)
	2	40	68.0 (32–85)	22 (55.0)
	3	25	68.0 (41–85)	15 (60.0)
**CP**	1	15	55.0 (37–75)	9 (60.0)
	2	35	52.0 (36–68)	21 (60.0)
	3	24	53.5 (36–68)	16 (66.7)
**GID**	1	0	0	0
	2	50	63.5 (25–83)	33 (66.0)
	3	26	63.0 (43–81))	19 (73.1)
**AIP-1**	1	8	56.5 (29–76)	6 (75.0)
	2	50	68.0 (29–84)	38 (76.0)
	3	47	65.0 (29–83)	35 (74.5)
**AIP-2**	1	7	43.0 (37–67)	5 (71.4)
	2	15	45.0 (32–76)	11 (73.3)
	3	15	45.0 (32–76)	11 (73.3)
**All patients**	1	60	63.0 (25–77)	33 (55.0)
	2	260	58.0 (20–85)	154 (59.2)
	3	185	59.0 (24–85)	127 (68.6)

**Table 2 ijms-21-02403-t002:** Characterization of the selected 29 autoantigens for individual screening.

Gene Symbol	Antigen Description	ORF Length (bp)
**CCKBR**	Cholecystokinin B receptor	1341
**CRYZL1**	Crystallin, zeta (quinone reductase)-like 1	1050
**TRUB1**	TruB pseudouridine (psi) synthase homolog 1 (*Escherichia coli*)	1047
**WDR45**	WD repeat domain 45	1083
**CYP3A5**	Cytochrome P450, family 3, subfamily A, polypeptide 5	1509
**IL13RA2**	Interleukin 13 receptor, alpha 2	1140
**ANXA4**	Annexin A4	963
**PAICS**	Phosphoribosylaminoimidazole carboxylase, phosphoribosylaminoimidazole succino-carboxamide synthase	1278
**EIF2S2**	Eukaryotic translation initiation factor 2, subunit 2 beta, 38 kDa	1002
**SLC7A7**	Solute carrier family 7 (cationic amino acid transporter, y+ system), member 7	1536
**RNF138**	Ring finger protein 138	738
**CNP**	2’,3’-cyclic nucleotide 3’ phosphodiesterase	1263
**AK1**	Adenylate kinase 1	585
**YTHDF2**	YTH domain family, member 2	1740
**ELF4**	E74-like factor 4 (ets domain transcription factor)	1992
**RAB31**	RAB31, member RAS oncogene family	585
**CHGA**	Chromogranin A (parathyroid secretory protein 1)	1374
**PSMC6**	Proteasome (prosome, macropain) 26S subunit, ATPase, 6	1167
**GPR3**	G protein-coupled receptor 3	993
**TOR1B**	Torsin family 1, member B (torsin B)	1011
**XRCC3**	X-ray repair complementing defective repair in Chinese hamster cells 3	1038
**ISOC1**	Isochorismatase domain containing 1	708
**LENG1**	Leukocyte receptor cluster (LRC) member 1	792
**PPA1**	Pyrophosphatase (inorganic) 1	870
**ZNF581**	Zinc finger protein 581	594
**PRSS1**	Protease, serine, 1 (trypsin 1)	720
**PPP1R15A**	Protein phosphatase 1, regulatory (inhibitor) subunit 15A	2025
**LTF**	Lactotransferrin	2136
**PRDX-4**	Peroxiredoxin 4	813

**Table 3 ijms-21-02403-t003:** Selected autoantigens and their normalized autoantibody levels (log2 of median fluorescence intensity) for all compared groups.

Gene Symbol	Co	PDAC	CP	GID	AIP-1	AIP-2
**CCKBR**	8.6305	8.7447	9.0325	8.6781	8.7038	8.9954
**CRYZL1**	8.7371	8.8043	8.9705	8.7381	8.8542	8.8528
**TRUB1**	8.7193	8.8316	9.1490	8.7414	8.7844	8.9571
**WDR45**	8.6397	8.7019	9.0841	8.8597	8.8556	8.8729
**CYP3A5**	8.6718	8.7179	9.0051	8.7582	8.7711	8.8270
**IL13RA2**	8.6985	8.7315	9.0586	8.7002	8.7162	8.9898
**ANXA4**	8.6938	8.7661	9.3695	8.7482	8.9397	8.8128
**PAICS**	8.6322	8.5264	9.2560	8.5043	9.0537	9.0789
**EIF2S2**	8.7413	8.6949	9.2072	8.6407	8.8949	8.7997
**SLC7A7**	8.6244	8.6772	9.1777	8.6259	8.7772	8.8422
**RNF138**	8.6440	8.6127	9.0573	8.5893	8.6385	9.0285
**CNP**	8.5802	8.5065	8.8401	8.4649	8.5370	8.9146
**AK1**	8.6401	8.6145	8.9979	8.5715	8.5416	8.9238
**YTHDF2**	8.4544	8.5643	8.8103	8.5429	8.4154	8.8491
**ELF4**	8.6501	8.6388	9.0962	8.7032	8.6739	8.8977
**RAB31**	8.6347	8.6322	9.0612	8.7034	8.7708	9.0932
**CHGA**	8.5721	8.5743	9.0147	8.5691	8.6180	8.6760
**PSMC6**	8.6986	8.7140	9.0575	8.6634	8.7957	8.7146
**GPR3**	8.7187	8.7172	9.0569	8.7065	8.7890	8.7467
**TOR1B**	8.6135	8.7880	9.1709	8.7872	8.8545	8.9956
**XRCC3**	8.6368	8.7289	9.1951	8.7563	8.7511	9.1535
**ISOC1**	8.6692	8.7422	9.0783	8.7196	8.7657	8.9872
**LENG1**	8.8537	8.8802	9.3380	8.8457	8.8833	8.8929
**PPA1**	8.9197	8.90522	9.5752	8.8729	9.1075	9.1996
**ZNF581**	8.9016	8.6838	9.2369	8.6845	9.0998	8.9404
**PRSS1**	8.4126	8.3032	8.9181	8.3791	8.4931	8.7592
**PPP1R15A**	8.8223	8.7876	9.5294	8.8394	9.1736	9.4495
**LTF**	8.5557	8.5745	8.906	8.4974	8.6029	8.4205
**PRDX-4**	7.9297	8.0862	8.7052	8.1987	8.0611	8.3418

**Table 4 ijms-21-02403-t004:** Autoantigens and their corresponding reactivity in PDAC and AIP patients’ sera. Selected autoantigens and their normalized autoantibody levels (log of median fluorescence intensity).

Gene Symbol	PDAC	AIP	*p* Value
**PAICS**	6091	6765	0.0436
**BOK**	6324	8439	0.0025
**RNF138**	4214	6352	0.0016
**TOR1B**	6727	10,077	5.56 × 10^−6^
**PPP1R15A**	5576	6458	0.0017
**LENG1**	5560	6391	0.0016
**CYP3A5**	5141	6717	0.0075
**GPR3**	4970	6481	0.0137
**CA2**	5576	4667	0.0173
**HDAC3**	4214	6352	0.0016

**Table 5 ijms-21-02403-t005:** Patients’ clinico-pathological parameters.

Variables	Patients (*n*)
**Total**	**315**
**Pancreatic cancer (PDAC)**	**65**
Age (years)	
Mean ± SD	66.2 ± 11
Range (median)	32–85 (68)
Gender: male/female	32/33
Grade	
**G1**	0
- **G2**	32
- **G3**	33
*TNM-stage* (AJCC stage, 8th edition)	
- T1a	1
- T1c	10
- T2	34
- T3	14
- T3 *	6
*N-stage*	
- N0	11
- N1	26
- N1*	3
- N2	25
*M-stage*	
- M0	53
- M1	12
*AJCC Stage*	
- IA	5
- IB	6
- IIA	2
- IIB	19
- IIB *	3
- III	18
- IV	12
*Tumour location*	
- head	42
- body	8
- tail	15
**Autoimmune pancreatitis (AIP)**	**70**
**AIP type 1**	**55**
Age (years)	
Mean ±SD	64.47 ± 12.21
Range (median)	29–84 (68)
Gender: male/female	42/13
**AIP type 2**	**15**
Age (years)	
Mean ±SD	51.86 ± 14.7
Range (median)	32–76 (44.5)
Gender: male/female	10/5
**Chronic pancreatitis (CP)**	**50**
Age (years)	
Mean ±SD	53.34 ± 9.6
Range (median)	36–75 (53.5)
Gender: male/female	33/17
**Gastrointestinal diseases (GID)**	**60**
Benign diseases	11
Malign diseases	49
- gastro, liver, colon, renal, other	6, 6, 17, 4, 16
Age (years)	
Mean ±SD	63.28 ± 10.75
Range (median)	25–83 (63)
Gender: male/female	41/19
**Healthy controls (Co)**	**70**
Age (years)	
Mean ±SD	46.8 ± 19.47
Range (median)	20–83 (40.5)
Gender: male/female	48/22

* AJCC stage, 7th edition (AJCC stage, 8th edition not available).

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
