# Peer review of "Novel Autoantibody Signatures in Sera of Patients with Pancreatic Cancer, Chronic Pancreatitis and Autoimmune Pancreatitis: A Protein Microarray Profiling Approach"

_ijms, 2020, doi:10.3390/ijms21072403_

Round 1

Reviewer 1 Report

I do not see any significant problems with this submission; recommend publication.

The paper is important since it has the potential to differentiate
between neoplastic and non-neoplastic pancreatic disease,
the latter of which can be treated, potentially, without surgical
intervention, thus preventing unnecessary "false positive" surgeries.
The scientific methodology is sound to identify targets for
antibody-mediated detection.
Issues of sensitivity and specificity are mentioned
in the Discussion. If there is any further suggestion,
it is that discussion could be expanded, since the consequence
of a "false negative" in this example is high.

Author Response

please see the attachment below, thanks.

Reviewer 2 Report

This paper tried to provide a fast and cost-effective way for pancreatic disease diagnosis via detecting autoantibody in sera. The study based on analyzing the amount of autoantibodies in healthy people and patients. Overall, the paper is okay but there are few questions need to be answered/amended before it can further consider for publication.

Major/Minor points:

  1. Actually, I think the authors need to explain more about those figures. For example, in Figure 2, it seems the authors compared a lot of thing among those different groups. There must have some information thus the authors need to explain/clarify clearly, and emphasize the novel autoantibody that the authors think it is significant.
  2. In Figure 1C, I can't fully understand the meaning of IQR in the table cell Median age. Besides, when the authors collected the information of patients in Figure 1C and Table 1, the authors only showed the population of male, thus is there any difference between male and female?
  3. Figure 2 and Figure 4 are too obscure to distinguish those letters.

Author Response

(The authors gave the same response as above.)

Round 2

Reviewer 2 Report

The authors have revised the manuscript accordingly. Thanks!